# Biosynthesis and Chemical Synthesis of Albomycin Nucleoside Antibiotics

**DOI:** 10.3390/antibiotics11040438

**Published:** 2022-03-24

**Authors:** Meiyan Wang, Yuxin Zhang, Lanxin Lv, Dekun Kong, Guoqing Niu

**Affiliations:** 1Biotechnology Research Center, Southwest University, Chongqing 400715, China; swuwmy@email.swu.edu.cn (M.W.); zhangyx950@email.swu.edu.cn (Y.Z.); l188165@email.swu.edu.cn (L.L.); kongdekun@swu.edu.cn (D.K.); 2Chongqing Key Laboratory of Plant Resource Conservation and Germplasm Innovation, Southwest University, Chongqing 400715, China; 3State Cultivation Base of Crop Stress Biology for Southern Mountainous Land, Academy of Agricultural Sciences, Southwest University, Chongqing 400715, China

**Keywords:** antibiotic resistance, albomycin, mode of action, biosynthesis, self-resistance, chemical synthesis, antimicrobial agents

## Abstract

The widespread emergence of antibiotic-resistant bacteria highlights the urgent need for new antimicrobial agents. Albomycins are a group of naturally occurring sideromycins with a thionucleoside antibiotic conjugated to a ferrichrome-type siderophore. The siderophore moiety serves as a vehicle to deliver albomycins into bacterial cells via a “Trojan horse” strategy. Albomycins function as specific inhibitors of seryl-tRNA synthetases and exhibit potent antimicrobial activities against both Gram-negative and Gram-positive bacteria, including many clinical pathogens. These distinctive features make albomycins promising drug candidates for the treatment of various bacterial infections, especially those caused by multidrug-resistant pathogens. We herein summarize findings on the discovery and structure elucidation, mechanism of action, biosynthesis and immunity, and chemical synthesis of albomcyins, with special focus on recent advances in the biosynthesis and chemical synthesis over the past decade (2012–2022). A thorough understanding of the biosynthetic pathway provides the basis for pathway engineering and combinatorial biosynthesis to create new albomycin analogues. Chemical synthesis of natural congeners and their synthetic analogues will be useful for systematic structure–activity relationship (SAR) studies, and thereby assist the design of novel albomycin-derived antimicrobial agents.

## 1. Introduction

The emergence and rapid spread of antibiotic resistance among pathogens has become a major healthcare problem worldwide. This problem becomes even worse with a coincident decline in the supply of novel antimicrobial agents [1,2]. Therefore, there is an urgent need for the discovery and development of novel antimicrobial drugs that either act on new cellular targets or bypass the development of resistance. Natural products offer a great variety of chemical diversity with distinctive biological activities. It is estimated that more than half of approved drugs, from the period of 1981 to 2019, are either natural products or derivatives thereof [3]. Thus, natural products remain the most promising sources for drug discovery and development [4,5]. In recent years, aminoacyl-tRNA synthetases (aaRSs) have emerged as attractive targets for the development of novel antimicrobial drugs [6]. To ensure the fidelity of protein synthesis, these enzymes catalyze the charging of tRNA with cognate amino acids in a two-step process [7]. The first step involves formation of an aminoacyl–adenylate (aa-AMP) intermediate with concomitant release of pyrophosphate (PP_i_). In the second step, the aminoacyl moiety is transferred to the 3′-terminal adenosine of the cognate tRNA (Figure 1). The 20 aaRSs have been classified into two classes: Class Ⅰ and Class Ⅱ, mainly based on the structures of their active sites [7]. Considering the vital role of aaRSs in protein synthesis, it is reasonable to expect that inhibition of these enzymes is detrimental to cell survival. 

Sideromycins are a unique subset of siderophores that are comprised of an antibacterial moiety covalently linked to a siderophore. They are actively transported into bacterial cells via a “Trojan horse” strategy by hijacking the siderophore uptake pathway, that is commonly used by bacteria to scavenge environmental iron [8]. These siderophore-antibiotic conjugates are promising drug candidates for the treatment of various bacterial infections, including those caused by multidrug-resistant pathogens. Over the past few decades, only a few naturally occurring sideromycins, such as albomycins, salmycins, and ferrimycins, have been identified [8,9]. Of particular interest are albomycins that function as inhibitors of bacterial seryl-tRNA synthetase (SerRS). Examples of other aaRSs inhibitors include microcin C and agrocin 84 (Figure 1), targeting aspartyl tRNA synthetase (AspRS) and leucyl-tRNA synthetase (LeuRS), respectively [10]. The biosynthesis, mode of action and self-resistance of microcin C have been reviewed comprehensively [10,11]. At present, our understanding on synthesis, mode of action and self-resistance of agrocin 84 is still limited [12,13]. In this review, we summarize findings on the discovery and structure elucidation, mode of action, biosynthesis, self-resistance, and chemical synthesis of albomycins.

## 2. Discovery and Structure Elucidation

Albomycin, originally designated as grisein, was first isolated from the soil-dwelling *Streptomyces griseus* in the 1940s [14,15]. It was also identified in *Streptomyces subtropicus* (previously known as *Actinomyces subtropicus*) [16,17]. It is noteworthy that albomycin was also known as alveomycin, antibiotics A 1787, LA 5352 and LA 5937, and Ro 5-2667 in the literature [18,19]. Thirty-five years after the initial isolation, chemical structures of three albomycin congeners (δ_1_, δ_2_, and ε) were fully elucidated by Benz and coworkers in 1982 [20,21]. The albomycins are composed of a ferrichrome-type siderophore and a 6′-amino-4′-thioheptose nucleoside that are linked via amide linkages to a serine residue (Figure 1). The siderophore moiety consists of three tandem *N*^5^-acetyl-*N*^5^-hydroxy-L-ornithine residues, mimicking the siderophore ferrichrome from fungi [22,23]. The most striking feature of the nucleoside moiety is the substitution of a sulfur atom for an oxygen atom on the pentose ring, that is most commonly found in other nucleoside antibiotics, such as liposidomycin, caprazamycin, muraymycin, muraminomicin, and A-90289 [24,25]. It is worth noting that the three albomycin congeners differ mainly in the C_4_ substituent of the pyrimidine nucleoside (Figure 1).

## 3. Mechanism of Action

Albomycins have attracted significant attention due to their potent antibacterial activities against both Gram-negative and Gram-positive bacteria [15,16,26]. For example, the major congener δ_2_ exhibited minimum inhibitory concentrations (MICs) as low as 5 ng/mL against *Escherichia coli* and 10 ng/mL against *Streptococcus pneumoniae* [27]. The excellent antibacterial activities of albomycins were attributed to their ability to hijack the ferric hydroxamate transport system and gain access to bacterial cells. For illustration purposes, we herein only briefly highlight current understanding on active uptake of albomycins in *E. coli* and *S. pneumoniae*. Readers interested in other bacteria species are referred to two recent reviews [10,28]. In *E. coli*, transport across the outer membrane is facilitated by a ferrichrome receptor FhuA [29], while transport across the cytoplasmic membrane is mediated by a ABC transporter FhuBCD [30]. FhuA consists of an N-terminal cork domain and a C-terminal transmembrane β-barrel domain. Binding of a ligand promotes the FhuA interaction with TonB, which in turn stimulates a rearrangement of the cork domain and the release of the ligand into the periplasmic space [31,32]. Once translocated into the periplasm, albomycin is bound by the periplasmic binding protein FhuD [33]. The antibiotic is then shuttled to the inner membrane ABC transporter FhuBC, and actively transported into the cytoplasm [34]. In *E. coli*, all genes encoding the transporter system are organized within the *fhuABCD* operon. Bacterial strains with mutations in any of these genes lost the ability to ingest albomycins and thus are resistant to their antimicrobial effects [35]. A similar transporter system was also characterized in *S. pneumoniae* [27]. The transporter system was encoded by the *fhu* locus consisting of *fhuD*, *fhuB*, *fhuG*, and *fhuC*. The *fhuD* encodes a binding lipoprotein, that is thought to be responsible for substrate recognition. The *fhuB* and *fhuG* encode transmembrane transport proteins, while the *fhuC* encodes an ATPase that presumably provides energy for transmembrane uptake [27]. Upon entry into the cell, albomycins will be hydrolyzed by host peptidases to release the thionucleoside warhead, referred to as SB-217452, from the iron-chelating siderophore moiety [36]. The SB-217452 resembles seryl adenylate, and selectively inhibits seryl-tRNA synthetases (SerRSs), and thereby interfering with host protein synthesis [27]. Thus, the potent antimicrobial activities of albomycins are attributed to the ferrichrome-mediated active uptake and subsequent liberation of the toxic thionucleoside SB-217452.

## 4. Biosynthetic Pathway

Albomycins are endowed with unique structural features, suggesting the occurrence of unusual enzymatic reactions during the biosynthetic process. To gain insight into this biosynthetic pathway, the gene cluster for albomycin biosynthesis has been identified by screening a cosmid library of *S. griseus* ATCC 700974 [37]. The gene cluster consists of 25 complete open reading frames (ORFs) including ORFs 1–7 and 18 genes from *abmA* to *abmR* (Figure 2A). It was proposed that the 18 genes from *abmA* to *abmR* are required for albomycin production, while ORFs 1–7 are not involved in albomycin biosynthesis [37]. A few recent studies have been directed to elucidate biosynthetic process of the thionucleoside SB-217452 [37,38,39]. These studies established that the formation of SB-217452 proceeds through complex enzymatic reactions with the participation of AbmH, AbmD, AbmF, AbmK, and AbmJ (Figure 2B). AbmH is a pyridoxal 5′-phosphate (PLP)-dependent transaldolase responsible for catalyzing a *threo*-selective aldol-type reaction to generate the thioheptose core with a D-ribofuranose ring and an L-amino acid moiety. Subsequently, the conversion of L- to D-amino acid configuration is achieved through the action of a PLP-dependent epimerase (AbmD) [38]. Then, an aminoacyl-tRNA synthetase (AbmF) catalyzes condensation between the 6′-amino-4′-thionucleoside and the D-ribo configuration and seryl-adenylate. Of note is that the seryl-adenylate appears to be supplied by the seryl-tRNA synthetase (AbmK). The D-*ribo* to D-*xylo* conversion of the thiofuranose ring is thought to be catalyzed by a radical S-adenosyl-L-methionine (SAM) enzyme (AbmJ) [38,39]. It is interesting to note that AbmK directly participates in the formation of SB-217452 by providing a substrate for albomycin biosynthesis. Similarly, a gene (*vlmL*) encoding a class II seryl-tRNA synthetase was also identified within the gene cluster for valanimycin biosynthesis in *Streptomyces viridifaciens*. Studies revealed that this seryl-tRNA synthetase is responsible for catalyzing seryl transfer in the valanimycin biosynthetic pathway [40,41]. Furthermore, two enzymes, a *N*-methyltransferase (AbmI) and a carbamoyltransferase (AbmE), have been identified to be responsible for the tailoring modifications of *N*^3^-methylation and *N*^4^-carbamoylation of cytidine [37]. Although significant progress has been made in uncovering biosynthetic machinery of SB-217452, the order of known steps needs further clarification. Moreover, many details of its biosynthesis remain elusive. For example, it remains a mystery how the characteristic sulfur atom is incorporated in **6**. Thus, more genetic and biochemical studies are required to reveal the complex process.

As mentioned above, the presence of the ferrichrome-type siderophore allows active transport of albomycin into bacterial cells. It was hypothesized that three genes, *abmA*, *abmB*, and *abmQ*, are responsible for the formation of the ferrichrome siderophore [37]. The pathway is initiated by the *N*^5^ hydroxylation of L-ornithine to *N*^5^-hydroxy-L-ornithine, catalyzed by a flavin-dependent ornithine monooxygenase (AbmB). Subsequently, *N*^5^-hydroxy-L-ornithine is converted to *N*^5^-acetyl-*N*^5^-hydroxy-L-ornithine through the action of an N-acyltransferase (AbmA). Three molecules of *N*^5^-acetyl-*N*^5^-hydroxy-L-ornithine are then used by, AbmQ, a nonribosomal peptide synthetase (NRPS), for iterative condensation to generate the tripeptide (Figure 2B). Sequence analysis suggested that AbmQ contains one adenylating (A) domain, two condensation (C) domains, and three thiolation (T) domains (Figure 2B). The presence of a single adenylating domain is a unique feature of AbmQ, implying that this domain supplies substrates for both condensation domains [37,39]. Furthermore, a typical NRPS contains a C-terminal thioesterase (TE) domain to catalyze hydrolysis of the thioester intermediate to release the peptide product from the phosphopantetheine group linked to the peptidyl carrier protein (PCP) domain [42]. However, AbmQ lacks a C-terminal TE domain. This atypical domain organization implies that no free siderophore is released from AbmQ and that the thioester intermediate may directly serve as the electrophile in the subsequent amide bond formation with either SB-217452 or L-serine. Genetic studies suggest that AbmC may participate in this amide bond formation [39]. However, biochemical evidence is needed to clarify its function. It is noteworthy that the biosynthetic pathway for the ferrichrome-type siderophore in albomycin has been proposed based on putative functions of associated genes. Further experimental evidence is definitely needed to support this assumption.

## 5. Self-Resistance

Typically, self-resistance is a prerequisite for antibiotic-producing bacteria. They have evolved a variety of mechanisms to ensure protection from self-made cytotoxic compounds. This is particularly true for the bacterial genus *Streptomyces*, prolific producers of antibiotics and many other bioactive secondary metabolites. The thionucleoside SB-217452 exhibits its antimicrobial activity by targeting SerRS. SerRS belongs to the class-II family of aminoacyl-tRNA synthetases (aaRSs) [43]. These enzymes catalyze the attachment of amino acids to their corresponding tRNAs [7]. Due to the pivotal role of aaRSs in the protein synthesis, inhibition of a member of this family is detrimental to cell viability. It is conceivable that the compound is also toxic to the natural producer. One early study identified two SerRS encoding genes (*serS1* and *serS2*) in the genome of albomycin-producing strain *S. griseus* ATCC 700974 [44]. Sequence analysis suggested that *serS1* encodes a housekeeping SerRS, while *serS2* encodes a SerRS that is significantly divergent from SerRS1. Of note is that *serS2* is located within the gene cluster for albomycin biosynthesis, thus it is designated as *abmK* [37]. However, *serS1* is situated elsewhere in the chromosome. When heterologously overexpressed in *E. coli* JM109, *serS2* (*abmK*) confers immunity to albomycin in vivo, while *serS1* fails to do so. Furthermore, SerRS2 was specifically resistant to SB-217452 in vitro [44]. A similar phenomenon has also been reported for other natural aaRSs inhibitors, such as mupirocin (isoleucyl-tRNA synthetases inhibitor) [45], borrelidin (threonyl-tRNA synthetase inhibitor) [46], and agrocin 84 (leucyl-tRNA synthetase inhibitor) [13].

It is not uncommon that antibiotic-producing bacteria duplicate genes encoding target proteins to confer resistance [45,47,48,49]. It is noteworthy that some resistance genes may be situated within or adjacent to the gene clusters for antibiotic biosynthesis. Based on these observations, resistance-guided genome mining strategy has been devised and used successfully for the discovery of novel antimicrobial agents [1,2,49,50]. Previous studies suggest that AbmK plays dual role in the biosynthesis and immunity of albomycins. A recent study indicated that AbmK participates directly in the formation of SB-217452 during biosynthesis of albomycins [39]. In the meantime, AbmK confers resistance to albomycins to avoid suicide [44]. Thus, *abmK* is a good candidate to be used as a query to search for homologues that will be helpful for the discovery of novel albomycin analogues or related compounds.

## 6. Chemical Synthesis

Chemical synthesis of the tri-δ-*N*-hydroxy-L-ornithine peptide siderophore was initially described by Benz and coworkers in 1984 [51,52]. Later, different synthetic strategies were described by Miller and coworkers in 1990s [53,54,55]. The synthesis of the thionucleoside moiety of albomycin δ_1_ (**13**) was also reported by Holzapfel and coworkers [56]. In one recent study, **13** and its isoleucyl and aspartyl analogues (**13a** and **13b**) were synthesized by Van Aerschot and coworkers [57]. Of note is that an oxygen analog of albomycin δ_1_ (**1d**) was synthesized by Benz and coworkers [58]. Surprisingly, the oxygen analog lost its antibacterial activity, suggesting that the sulfur atom is indispensable for the bioactivity of albomycins [59]. Furthermore, two aryl-tetrazole containing albomycin analogues (**15a** and **15b**) were obtained by Van Aerschot and coworkers (Figure 3). The two analogues were synthesized by coupling of aryl-tetrazole-containing compounds **14a** (CB168) and **14b** (CB432) to the tri-δ-*N*-hydroxy-L-ornithine peptide siderophore [60]. CB168 and CB432 were synthesized by Cubist Pharmaceuticals as potent aaRS inhibitors. Both CB168 and CB432 contain an aryl-tetrazole in place of the adenine moiety, and connected through a two-carbon linker to ribose [61]. The tri-δ-*N*-hydroxy-L-ornithine peptide siderophore were prepared as described by Miller and coworkers [53]. Antimicrobial activities were then tested against *Staphylococcus aureus* (ATCC 6538), *Staphylococcus epidermidis* RP62A (ATCC 35984), *Pseudomonas aeruginosa* PAO1, *Sarcina lutea* (ATCC 9341), and *Candida albicans* CO11. Only one of the compounds showed weak activity against *S. aureus* and *C. albicans*, though both analogues showed good activity against IleRS in vitro [60]. Total synthesis of the three natural albomycin congeners has been accomplished recently by He and coworkers [59]. For illustration purposes, we here only show a simplified scheme of albomycins synthesis (Figure 2). The efficient synthesis of albomycins enables biological evaluations of these potent inhibitors against three Gram-positive bacteria (*S. pneumoniae* ATCC 49619, *S. aureus* USA 300 NRS 384, and *Bacillus subtilis* ATCC 6633) and three Gram-negative bacteria (*E. coli* BJ 5183, *Neisseria gonorrhoeae* ATCC 49226, and *Salmonella typhi*), as well as 27 clinical *S. pneumoniae* and *S. aureus* isolates (three of them are methicillin-resistant strains). The results revealed that C_4_ substituent on the nucleobase in albomycin plays an essential role in their antibacterial activity. Importantly, the systematic SAR study suggested that albomycin δ_2_ is the most promising candidate for further clinical drug development. Of particular interest is that the potency of albomycin δ_2_ exceeds that of well-established antibiotics ciprofloxacin, vancomycin, and penicillin [59]. Inspired by these studies, more previously inaccessible analogues can be synthesized and subjected to systematic SAR studies. These studies will substantially expand the repertoire of novel albomycins that will ultimately expedite the design and development of effective antimicrobial agents.

## 7. Concluding Remarks

Aminoacyl-tRNA synthetases (aaRSs) catalyze the charging of tRNA with a cognate amino acid. They are indispensable for protein synthesis in all three kingdoms of life. Due to their important role in cellular viability, aaRSs have been recognized as suitable targets for the development of novel antimicrobial agents [6]. At present, there are three aaRS inhibitors in clinical practice. Mupirocin is an inhibitor of bacterial isoleucyl-tRNA synthetase (IleRS). It is mainly used as an antibacterial agent for the treatment of infection caused by Gram-positive bacteria, including methicillin-resistant *Staphylococcus aureus* (MRSA) [62,63]. Tavaborole is an inhibitor of fungal leucyl-tRNA synthetases (LeuRSs). It was approved by the US Food and Drug Administration (FDA) for the treatment of onychomycosis caused by *Trichophyton* species [64,65]. Halofuginone is an inhibitor of proline-tRNA synthetases (ProRSs). This drug was FDA approved for the treatment of apicomplexan parasite infections in chickens and scleroderma in humans [66].

Albomycins utilize the “Trojan horse” strategy to enter target bacterial cells. Once inside the cell, albomycins will be hydrolyzed by host peptidases to release the thionucleoside SB-217452 warhead that targets bacterial SerRSs. Thus, albomycins exhibit potent antibacterial activities against many model bacteria and a number of clinical pathogens [59]. Over the past several decades, a significant progress has been made on the mechanism of action, biosynthesis and immunity, and chemical synthesis of albomycins. Previous studies suggested that the biosynthetic pathway proceeds through multiple steps of unusual enzymatic reactions [38,39]. However, there are still many missing gaps in our understanding of the biosynthetic process. Therefore, further genetic and biochemical studies are needed to bridge these gaps. For example, further studies are needed to clarify the functions of AbmQ and AbmC in the amide bond formation. Moreover, further experimental evidence is also required for the formation of ferrichrome-type siderophore moiety. These studies will provide the basis for pathway engineering and combinatorial biosynthesis to create new albomycin analogues.

Natural products still hold out the best options for finding novel antimicrobial agents [3]. Advances in omics and bioinformatics have accelerated the identification of gene clusters for antibiotics biosynthesis. In recent years, resistance-guided genome mining has been used successfully for the discovery of novel antimicrobial agents [1,2,49,50]. As mentioned above, *abmK* can serve as a good candidate to search for homologous genes in public databases. Analysis of *abmK* homologues and the neighboring genes will be helpful for high-throughput screening of gene clusters encoding novel albomycin analogues. This large-scale genome mining open up new opportunities for the discovery and development of antimicrobial drugs to address the growing global threat of antimicrobial resistance.

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
