# Peer review of "Biosynthesis and Chemical Synthesis of Albomycin Nucleoside Antibiotics"

_antibiotics, 2022, doi:10.3390/antibiotics11040438_

Round 1

Reviewer 1 Report

Dear Editor,
MDPI-Antibiotics
I have evaluated the review article (Antibiotics-1640942) titled “Biosynthesis and Chemical Synthesis of Albomycin Nucleoside Antibiotics” by Niu et.al. I am delighted to review this review article, about recent advances in biosynthesis and chemical synthesis of Albomycin. The review does not follow the scope and limitation of the journal Antibiotics. This review lacks in-depth discussion with poor presentation.
I would recommend the article could be published in Antibiotics with major corrections. The authors need to address the below-mentioned queries.

1.    The author needs to mention the range of years covered in this review.
2.    The author needs to show the diagram for the enzymatic activity of aminoacyl-tRNA synthetases (aaRSs).
3.    Show the structure of “few naturally occurring sideromycins have been identified”.
4.    Although heating is ‘Discovery and structure elucidation”; however, the author didn’t discuss in detail structure elucidation.
5.    Pink color in Figure 1 is missing “Parts shaded in pink indicate the thionucleoside scaffold.”
6.    The author should show some figures for “Mechanism of action” instead of citing the previous review references.
7.    Each step of Figure 2 needs to be discussed in detail.
8.    Although the author mentioned that this review is going to cover a “special focus on recent advances in the biosynthesis and chemical synthesis” however, no chemical synthetic schemes were found in this review. The author only cited the references.
9.    The author unnecessarily repeated the known fact “Natural products still hold out the best options for finding novel antimicrobial agents”

Author Response

I have evaluated the review article (Antibiotics-1640942) titled “Biosynthesis and Chemical Synthesis of Albomycin Nucleoside Antibiotics” by Niu et.al. I am delighted to review this review article, about recent advances in biosynthesis and chemical synthesis of Albomycin. The review does not follow the scope and limitation of the journal Antibiotics. This review lacks in-depth discussion with poor presentation. I would recommend the article could be published in Antibiotics with major corrections. The authors need to address the below-mentioned queries.

1. The author needs to mention the range of years covered in this review.

Response: Thank you for the advice. The range of years covered was added in the Abstract.

2. The author needs to show the diagram for the enzymatic activity of aminoacyl-tRNA synthetases (aaRSs).

Response: This is a good suggestion. We added a new figure in the revised manuscript.

3. Show the structure of “few naturally occurring sideromycins have been identified”.

Response: The structures of salmycin and ferrimycin were provided as suggested.

4. Although heating is ‘Discovery and structure elucidation”; however, the author didn’t discuss in detail structure elucidation.

Response: As stated in our manuscript, the structures of albomycins were fully elucidated in 1982. In this review, we mainly focus on biosynthesis and chemical synthesis of albomycins. Therefore, we did not discuss structure elucidation in more details.

5. Pink color in Figure 1 is missing “Parts shaded in pink indicate the thionucleoside scaffold.”

Response: We are sorry for the confusion. We corrected it in the revised figure legend.

6. The author should show some figures for “Mechanism of action” instead of citing the previous review references.

Response: Figures have been shown for “Mechanism of action” in two recent reviews that we cited in our manuscript. To avoid repeating the same illustration, we think it is better to give a brief introduction of “Mechanism of action”.

7. Each step of Figure 2 needs to be discussed in detail.

Response: We think it is unnecessary to discuss each step of the biosynthetic pathway. We add more sentences of discussion in the revised manuscript.

8. Although the author mentioned that this review is going to cover a “special focus on recent advances in the biosynthesis and chemical synthesis” however, no chemical synthetic schemes were found in this review. The author only cited the references.

Response: We provided a chemical synthetic scheme in the revised manuscript.

9. The author unnecessarily repeated the known fact “Natural products still hold out the best options for finding novel antimicrobial agents”

Response: This sentence in Concluding Remarks” was used to usher in the following sentences. We would prefer to keep it.

Reviewer 2 Report

Dear Authors,

      I have reviewed the manuscript entitled "Biosynthesis and Chemical Synthesis of Albomycin Nucleoside Antibiotics" and found it to be very well written and of interest for the scientific community as it presents an update on a new class of antibiotics, the albomycin nucleoside antibiotics.

     There a few small observations to be taken into account by the authors prior to considering this manuscript for publication. They are mentioned below:

  1. Line 82: There's no pink in the figure representing the albomycin congeners. I believe "pink" should be corrected to "blue".
  2. Line 85: Please write "Gram-negative" instead of "Gram negative" as written in the rest of the manuscript.
  3. Line 87: Please correct "pneumonia" to "pneumoniae".
  4. Line 102: I recommend the following rephrasing: "in any of these genes".
  5. Line 121: Please correct "are not involve" to "are not involved".
  6. Line 135: I recommend the following rephrasing: "a gene (vlmL) encoding a class II".
  7. Line 157: Please correct "is the used" to "are then used".
  8. Line 168: Please correct "suggests" to "suggest".
  9. Line 197: I recommend the following changes: "may be situated within".
  10. Line 212: I suggest the following modification: "Of note is that an oxygen analog".
  11. Line 225: Please correct "compound" to "compounds".
  12. Line 226: I believe "though" would be more suitable in the context, instead of "through".
  13. Line 231: The ATCC code shoulb be specified for Salmonella typhi.
  14. Line 235: I suggest the following change: "is the most promising".
  15. Line 257: I believe "utilize" should be used instead of "utilized".

Author Response

I have reviewed the manuscript entitled "Biosynthesis and Chemical Synthesis of Albomycin Nucleoside Antibiotics" and found it to be very well written and of interest for the scientific community as it presents an update on a new class of antibiotics, the albomycin nucleoside antibiotics.

There a few small observations to be taken into account by the authors prior to considering this manuscript for publication. They are mentioned below:

  1. Line 82: There's no pink in the figure representing the albomycin congeners. I believe "pink" should be corrected to "blue".

Response: Yes. You are right. We corrected it.

  1. Line 85: Please write "Gram-negative" instead of "Gram negative" as written in the rest of the manuscript.

Response: We corrected it as suggested.

  1. Line 87: Please correct "pneumonia" to "pneumoniae".

Response: We corrected it.

  1. Line 102: I recommend the following rephrasing: "in any of these genes".

Response: We changed it as suggested.

  1. Line 121: Please correct "are not involve" to "are not involved".

Response: We corrected it.

  1. Line 135: I recommend the following rephrasing: "a gene (vlmL) encoding a class II".

Response: We changed it as suggested.

  1. Line 157: Please correct "is the used" to "are then used".

Response: We corrected it.

  1. Line 168: Please correct "suggests" to "suggest".

Response: We corrected it.

  1. Line 197: I recommend the following changes: "may be situated within".

Response: We changed it as suggested.

  1. Line 212: I suggest the following modification: "Of note is that an oxygen analog".

Response: We changed it as suggested.

  1. Line 225: Please correct "compound" to "compounds".

Response: We corrected it.

  1. Line 226: I believe "though" would be more suitable in the context, instead of "through".

Response: We corrected it.

  1. Line 231: The ATCC code should be specified for Salmonella typhi.

Response: We checked the original reference. There is no ATCC number specified for this strain.

  1. Line 235: I suggest the following change: "is the most promising".

Response: We changed it as suggested.

  1. Line 257: I believe "utilize" should be used instead of "utilized".

Response: We corrected it.

Reviewer 3 Report

This paper describes the biosynthesis and the chemical synthesis of albomycin nucleoside-based antibiotics that utilized a "Trojan Horse" mechanism to target bacterial cells. The work has been carried out pretty well and has been described in a lucid manner. I fail to see any major discrepancies or shortcomings in this article. Therefore, I recommend its publication in Antibiotics. 

I have some general questions and would recommend the authors consider them adding in the revised version. 

1) I realize that this article focused on natural product-based antibiotics. I was wondering whether the authors find any example where this "Trojan Horse" approach has been utilized but with a synthetic compound (even say a metal complex). If there are such examples those should be included in the Introduction section with relevant references. 

2) Also along the same line as question 1, are there any examples where this approach was used to deliver other therapeutics (other than antibacterial agents), if yes, the authors should also mention those examples with a few lines (including the relevant references). 

Author Response

This paper describes the biosynthesis and the chemical synthesis of albomycin nucleoside-based antibiotics that utilized a "Trojan Horse" mechanism to target bacterial cells. The work has been carried out pretty well and has been described in a lucid manner. I fail to see any major discrepancies or shortcomings in this article. Therefore, I recommend its publication in Antibiotics. 

I have some general questions and would recommend the authors consider them adding in the revised version. 

1) I realize that this article focused on natural product-based antibiotics. I was wondering whether the authors find any example where this "Trojan Horse" approach has been utilized but with a synthetic compound (even say a metal complex). If there are such examples those should be included in the Introduction section with relevant references. 

Response: Thank you for the advice. There are synthetic compounds with "Trojan Horse" features. For example, Cefiderocol is a siderophore cephalosporin antibiotic. Cephalosporin belongs to β-lactam antibiotics. It inhibits bacterial cell wall synthesis by binding to penicillin binding proteins. In this review, we tried to focus on aminoacyl-tRNA synthetases inhibitors, such as albomycins. Thus, we did not include these compounds in our manuscript.

2) Also along the same line as question 1, are there any examples where this approach was used to deliver other therapeutics (other than antibacterial agents), if yes, the authors should also mention those examples with a few lines (including the relevant references). 

Response: We are sorry that we did not notice this approach used for other therapeutics.

Round 2

Reviewer 1 Report

I would like to thank the authors of the manuscript for the consideration of the reviews’ comments and for making the necessary corrections/changes accordingly. The author has made necessary corrections and modifications to the manuscript as suggested by the reviewers which remarkably enhanced the overall quality of the manuscript with a clear presentation of results and proper discussion. 

Some recommendations of the journal for manuscript presentation are ignored but I believe that the editor will point out these problems.

I would recommend the article could be published in Antibiotics in the present form, however, the author could consider the following minor corrections

  1. Number the compounds in Figure 1.
  2. Better use term Scheme for Figure 1, Figure 3, and Figure 5.
  3. For Figure 5: Just mention the number of steps represented by the double arrows and write the missing reagent in the arrows if possible or the name of the reactions involved.
  4. The author could include the following reference

(i) Shiraishi, T., Kuzuyama, T. Recent advances in the biosynthesis of nucleoside antibiotics. J Antibiot 72, 913–923 (2019). https://doi.org/10.1038/s41429-019-0236-2

Author Response

We would like to thank you again for comments on our manuscript. Your suggestions are of great help to us. Our responses to the individual points are as follows.  

1. Number the compounds in Figure 1.

Response: We made changes as suggested.

2. Better use term Scheme for Figure 1, Figure 3, and Figure 5.

Response: We changes Figure 1 and Figure 5 to Scheme 1 and 2, respectively. For Figure 3, we think it is better to keep it as a Figure. This figure contained both genetic organization of the gene cluster and the biosynthetic pathway. If only the biosynthetic pathway was included, it is better to use scheme.  

3. For Figure 5: Just mention the number of steps represented by the double arrows and write the missing reagent in the arrows if possible or the name of the reactions involved.

Response: Thank you for the advice. We made changes as suggested.

4. The author could include the following reference (i) Shiraishi, T., Kuzuyama, T. Recent advances in the biosynthesis of nucleoside antibiotics. J Antibiot 72, 913–923 (2019). https://doi.org/10.1038/s41429-019-0236-2

Response: This is a good suggestion. The reference was included as ref. 25.